# An optimization tool for identifying Multiple Diffusion Domain Model parameters

Andrew L. Gorin[1,2], Joshua M. Gorin[3], Marie Bergelin[2], and David L. Shuster[1,2]

[1]Department of Earth and Planetary Science, University of California, Berkeley, CA
[2]Berkeley Geochronology Center, Berkeley, CA
[3]Independent Researcher

**Correspondence:** Andrew Gorin (andrew_gorin@berkeley.edu)

**Abstract.** The Multiple-Diffusion Domain (MDD) model empirically describes the diffusive behavior of noble gases in some terrestrial materials and has been commonly used to interpret $^{40}$Ar/$^{39}$Ar stepwise degassing observations in K-feldspar. When applied in this manner, the MDD model can be used to test crustal exhumation scenarios by identifying the permissible thermal paths a rock sample could have undergone over geological time, assuming the diffusive properties of Ar within the mineral are accurately understood. More generally, the MDD model provides a framework for quantifying the temperature-dependent diffusivity of noble gasses in minerals. However, constraining MDD parameters that successfully predict the results of step-heating diffusion experiments is a complex task and the assumptions made by existing numerical methods used to quantify model parameters can bias the absolute temperatures permitted by thermal modeling. For example, the most commonly used method assumes that no domains lose more than 60% of their gas during early heating steps [Lovera et al. 1997, Geochimica et Cosmochimica Acta, 61, 3171–3192]. This assumption is unverifiable, and we show that Lovera et al.'s (1997) procedure may bias predicted temperatures towards lower values when it is violated. To address this potential bias and to provide greater accessibility to the MDD model, we present a new, open-source method for constraining MDD parameters from stepwise degassing experimental results, called the "MDD Tool Kit." This software optimizes all MDD parameters simultaneously and removes any need for user-defined $E_a$ or regression-fitting choices used by other tools. In doing so, this new method eliminates assumptions about the domain size distribution. To test the validity of our thermal predictions, we then use the MDD Tool Kit to interpret $^{40}$Ar/$^{39}$Ar results from the Grayback Fault, AZ, USA. Although the resulting thermal histories are consistently $\sim$ 60 – 75 °C higher than those found in previous studies, they agree with independent observations from apatite fission track, zircon fission track, and (U-Th)/He.

## 1 Introduction

$^{40}$Ar/$^{39}$Ar thermochronology is a valuable tool for studying Earth's crustal exhumation because it constrains a mineral's continuous thermal history through geologic time (McDougall and Harrison, 1999). While $^{40}$Ar/$^{39}$Ar geochronology was initially developed to quantify crystallization timing of rapidly cooled igneous rocks (Turner, 1968), its application to thermochronology did not begin in earnest until Dodson's (1973) quantitative description of closure theory, which allowed one to determine when a mineral cooled below a mineral-specific "closure temperature" for radiogenic $^{40}$Ar retention (Reiners, 2005). Dur-

ing this early period of thermochronology research, minerals and isotopic systems with disparate closure temperatures were combined to constrain a given rock's permissible thermal histories (e.g. Wagner, 1977). While this approach yielded valuable insight into Earth's surface processes (McDougall and Harrison, 1999), it requires that multiple mineral phases—which are not always present—to coexist in a sample. Further, the thermal histories inferred from this technique are discontinuous, with one age-temperature measurement for each mineral phase, thus limiting the ability to quantify cooling rates at specific times (McDougall and Harrison, 1999), and potentially introducing biases. Additionally, this approach does not fully accommodate the complexity of the diffusive properties observed in some minerals (Zeitler, 1987; McDougall and Harrison, 1999).

Since the late 1960s, it had been understood that complex retention properties pose challenges for interpreting the thermal histories of some samples. More specifically, the results of some stepwise degassing experiments were not consistent with volume diffusion from a single domain in some minerals, requiring a new interpretative framework (Zeitler, 1987; Lovera et al., 1989). Progress came first with Zeitler's (1987) observations in K-feldspar that the anomalous behavior seen in their $^{40}$Ar/$^{39}$Ar age spectra and associated Arrhenius plots could be explained by outward diffusion of $^{39}$Ar from several "domains" (distinct, non-interacting geometries in which volume diffusion takes place) of varying sizes simultaneously. A formalism for this model, called the Multiple-Diffusion Domain (MDD) Model, was then described by Lovera et al (1989). This model enabled more nuanced interpretations of sample-specific diffusion kinetics—and thereby thermal histories—of minerals.

While this model has received criticism (e.g. Villa, 1994; Parsons et al., 1999; Popov and Spikings, 2020; Popov et al., 2020a, b; Spikings and Popov, 2021), it has largely been adopted by the thermochronology community since the 2000s due to its ability to constrain time-temperature (t-T) histories that are consistent with both an observed $^{40}$Ar/$^{39}$Ar age spectrum and calculated diffusion kinetics of a given sample (Reiners, 2005; Harrison and Lovera, 2014). However, we show that the method used by Lovera et al. (1997) to tune the MDD model parameters to match the results of a step-heating diffusion experiment potentially introduces a systematic bias towards colder temperatures through geologic time. This was recognized, but not addressed, at the time of their publication, likely due to insufficient computing power to resolve the issue (Lovera et al., 1997).

## 2  The MDD Model

The MDD model is best understood through analogy to volume diffusion through a single domain. In this simple case, the temperature-dependent diffusivity is described by

$$\frac{D(T)}{a^2} = \frac{D_0}{a^2} * e^{\frac{-E_a}{RT}} \qquad (1)$$

where $T$ is the absolute temperature (K), $D$ is diffusivity at $T$ in (cm$^2\cdot$ sec$^{-1}$), $D_0$ is diffusivity at infinite $T$, $E_a$ is activation energy (kJ·mol$^{-1}$), $R$ is the gas constant (kJ·mol$^{-1}$·K$^{-1}$), and $a$ is the radius of the diffusion domain (cm) (see Table 1 for variable definitions).

This relationship is typically determined empirically for noble gas diffusion in minerals through a stepwise degassing diffusion experiment. In such experiments, the diffusant is first produced in-situ by a proton or neutron irradiation to ensure a

homogeneous initial distribution. The sample is then placed under static vacuum where it is repeatedly heated to a known temperature, and the quantity of the diffusant is measured (Fechtig and Kalbitzer, 1966; McDougall and Harrison, 1999). By assuming a geometry for the diffusion domain, the fractional loss at each step can be used to calculate a corresponding $\frac{D}{a^2}$

with the equations of Fechtig and Kalbitzer (1966) or Crank (1975). Ginster and Reiners (2018) summarized and propagated measurement uncertainties through these equations and we use their forms in this work (Table 2).

When consistent with volume diffusion through a single diffusion domain, a step-heating experiment will produce a linear Arrhenius relationship between calculated values of $\log(\frac{D}{a^2})$ and $\frac{1}{T}$. In these cases, a linear regression can then be fit to the results to determine $E_a$ and $\frac{D_0}{a_0^2}$ (Lovera et al., 1997) for a sample. Several mineral-diffusant pairs such as $^3$He in apatite and

$^3$He in olivine exhibit such behavior (Shuster et al., 2004).

However, not all minerals exhibit diffusive behavior consistent with volume diffusion from a single diffusion domain. The diffusive behavior of some minerals appears to be more consistent with diffusion from several, non-interacting domains of varying sizes, within a given mineral, diffusing simultaneously. This behavior was later identified and roughly quantified by Gillespie et al. (1982) and Zeitler (1987) in K-feldspar, orthoclase, and microcline, and then formalized by Lovera et al. (1989).

In contrast to the single diffusion domain model, the MDD model can be imagined as a series of non-interacting diffusion domains of varying diffusive length scale (e.g., different radii, $a$) that all diffuse simultaneously. The choice of geometry for the domains is somewhat arbitrary, as it remains unclear what these diffusion domains physically represent in a mineral sample (Harrison and Lovera, 2014; Parsons et al., 1999). Some authors use a plane sheet, while others prefer a spherical or cylindrical geometry. Regardless of this choice, it is assumed that each diffusion domain has the same geometry, $E_a$, and $D_0$. Therefore,

a diffusion domain can be described completely by three parameters: (i) $E_a$ (common to all domains), (ii) its pre-exponential term $(\frac{D_0}{a^2})_i$ and (iii) the proportion of the total gas it contains $(\phi)$. Thus, a multiple-diffusion domain model with n domains has $2n-1$ free parameters because $\sum_{i=1}^{n} \phi_n = 1$.

An inherent challenge in applying this model is that each of these $2n-1$ parameters need to be optimized to accurately predict the results of a given diffusion experiment. Because authors using the MDD model fit as many as 10 domains, this

optimization is regularly performed on 19-dimensional vectors or larger, making the exercise non-trivial.

## 2.1 Critiques of the MDD Model

Validating the MDD model's assumptions is beyond this paper's scope; here we briefly review the assumptions and their prior critiques. The foundational assumption of the MDD model is that the transport of $^{40}$Ar within minerals over geologic time occurs primarily by volume diffusion. This assumption predicts that low $^{40}$Ar concentrations should exist near the outer

edge of a crystal (i.e., for a boundary condition of nearly zero $^{40}$Ar concentration external to the crystal), and that higher concentrations should exist towards the mineral interior, translating to apparently younger and older $^{40}$Ar/$^{39}$Ar ages in those locations, respectively. Studies have documented such spatial correlations (Flude et al., 2014). However, others have shown that polyphasic samples can violate this expectation (Popov and Spikings, 2020); such minerals are not suitable candidates for MDD modeling.

It is also assumed that diffusive behavior observed during laboratory step-heating experiments is the same as under natural conditions over geologic time. The MDD model proposes that the diffusive behavior of Ar within some minerals can be described by numerous, non-interacting, infinite sheets simultaneously diffusing within the same mineral. As evidence, Lovera et al. (2002) argued that a correlation between the $^{40}$Ar/$^{39}$Ar age spectra and log(r/r0) plots (see Lovera et al. (2002), for description of log(r/r0) plots) validates, or is at least consistent with, these assumptions. They find, however, that only $\sim 40\%$ of

K-feldspars demonstrate sufficient correlation for MDD modeling and suggest that the remaining samples have been affected by recrystallization or other mineral inclusions (Lovera et al., 1997). Other authors have gone further, asserting that recrystallization within minerals is nearly always responsible for this behavior (Popov and Spikings, 2020; Popov et al., 2020a, b; Spikings and Popov, 2021). While complex mineralogy is likely responsible in some cases, detailed petrologic examinations and a strong correlation between an $^{40}$Ar/$^{39}$Ar age spectrum and a log(r/r0) plot can increase confidence that the MDD model

framework is applicable to an individual sample. Indeed, such petrologic investigations are commonplace in MDD studies of K-feldspars (e.g. Wong et al., 2023).

    Others have questioned the primary assumption that Ar diffusion under laboratory conditions functions the same way as natural diffusion over geologic time by suggesting that mineral structures may be altered during diffusion experiments. Popov et al., (2020a) observed the development of cracks within crystals during some experiments. However, Lovera et al. (1993)

performed a double-irradiation experiment where they measured the diffusion kinetics of a grain up to $\sim 850$ °C, reirradiated it and then successfully reproduced the diffusion kinetics of the first experiment, indicating that cracking or other annealing effects were not a primary cause of MDD behavior in that sample. The Lovera et al. (1993) study also indicated that the diffusion kinetics of the mineral were not altered by heating during the diffusion experiment.

    Some further MDD-model assumptions are made which require knowledge of the physical structure of the domains. For

example, it is assumed that Ar is instantaneously removed from the mineral after reaching a domain boundary, that diffusion in each domain is isotropic, that the domains are all formed at the same time, and that the parent isotope, $^{40}$K, is uniformly distributed within each domain (Parsons et al., 1999; Lovera et al., 1989). Because it is unclear what the diffusion domains physically represent within a mineral, these assumptions are currently unverifiable. Although it is unsatisfying that a physical representation of the diffusion domains is not clearly identifiable within K-feldspar and other minerals, the MDD model need

not be rejected as a tool for thermochronology. Indeed, ample evidence suggests that the MDD model reliably predicts thermal histories supported by independent thermochronological data (e.g. Warnock and Zeitler, 1998; Reiners and Farley, 1999; Axen et al., 2000; Spell et al., 2000; Kirby et al., 2002; Reiners et al., 2004; Shirvell et al., 2009). Though empirical, the MDD model remains a useful tool for constraining such histories through geologic time.

## 3   Existing Optimization Methods

The Lovera et al. (1997) method for identifying MDD model parameters is the only published algorithm for this purpose that we are aware of and has been used in many studies (e.g. Harrison et al., 1995; Quidelleur et al., 1997; Grove et al., 2003; Weirich et al., 2012; Wong et al., 2023). The routine begins by defining a reference domain which is used to calculate an $E_a$ by

fitting an uncertainty-weighted linear regression to the $\log_{10}(\frac{D}{a^2})$ values resulting from a subset of the low temperature heating steps in an Arrhenius plot corrected for excess Ar released from fluid inclusions (Harrison et al., 1994); the resultant $E_a$ is assumed applicable to all domains. This is done because—even in samples exhibiting MDD-like behavior—these diffusivities tend to approximate a line (Figure 1A).

To determine the number of points to include in the uncertainty-weighted linear regression, the lowest temperature heating steps are sequentially added to the regression. After adding each step, a chi-squared misfit static is calculated and is then used to calculate a goodness-of-fit probability ($q$) (Lovera et al., 1997). These relationships are described as follows:

$$\chi^2_{lovera} = \sum_{i=1}^{N} (\frac{\log_{10}(\frac{D}{a^2})_i - \log_{10}(\frac{\hat{D}}{a^2})_i}{\sigma_i})^2 \tag{2}$$

$$q = \text{gammq}(\frac{N-2}{2}, \frac{\chi^2}{2}) \tag{3}$$

where $gammq$ is the incomplete gamma function, $(\frac{D}{a^2})_i$ is the observed pre-exponential term, $(\frac{\hat{D}}{a^2})_i$ is the modeled pre-exponential term, $\sigma$ is the uncertainty on the observed pre-exponential term, and $N$ is the number of steps included (Lovera et al., 1997; Press, 2007). This calculation is repeated until the value of $q \cdot N$ is maximized. Once the $E_a$ is determined from the above process, this value is held fixed while a Levenberg-Marquardt method (Press, 2007) is used to adjust the gas fraction ($\phi$) and $\log_{10}(\frac{D_0}{a^2})$ for each domain to minimize the above chi-square quantity. The reader is directed to Appendix B of Lovera et al. (1997) and the referenced sections of Press (2007) for additional information.

While this routine is generally robust and capable of accurately quantifying $E_a$ in synthetic data experiments where the kinetics are defined, it fails to do so when the smallest domains lose greater than 60% of their gas in early heating steps (Figure 1). Thus, use of this routine makes the implicit assumption that no domains diffuse most of their gas during the initial heating steps. When this assumption is violated, the regression will overestimate the $E_a$ and $\log_{10}(\frac{D_0}{a_0^2})$ (Figure 1). Because there is currently no way to know the domain size distribution *a priori*, it is not possible to know whether this assumption is valid for any given sample (Lovera et al., 1997).

Furthermore, Lovera et al. (1997) found evidence that the variability of predicted $E_a$ values from different aliquots of the same mineral decreased when they were able to use a higher percentage of the total gas released in their linear regressions. Simply put, the more gas included in the calculation, the less variation there was in predicted $E_a$. This observation suggests that a routine which maximizes the fraction of $^{39}$Ar used in the fitting exercise might lead to more precise estimates of diffusion kinetics.

## 4 The MDD Tool Kit Approach

With the increased computational power since the publication of the original routine for optimizing these models, we apply SciPy's implementation of the differential evolution algorithm to fit all the diffusion kinetics parameters simultaneously (Storn

and Price, 1997; Lampinen, 2002; Qiang and Mitchell, 2014; Virtanen et al., 2020). This algorithm is a population-based genetic algorithm for optimizing over continuous search spaces in high dimensions with non-linear and non-differentiable misfit functions (Storn and Price, 1997).

## 4.1 Differential Evolution

Our implementation of SciPy's differential evolution is a method for iteratively improving a randomly selected "population" of guesses until the best-fitting diffusion kinetics are found (Figure 2). The algorithm improves these guesses by combining elements of the most successful vectors with those remaining. In this case, success refers to the value of a misfit statistic calculated between the guess's forward-modeled gas releases and the observed results. Through this process, each successive generation achieves a misfit equal to or lower than the previous one.

We apply two such misfit statistics to the data analyzed in this study, one error weighted, $\chi^2$, and another where all points are weighted equally ($\%_{frac}$). Our $\chi^2$ misfit accounts for the uncertainty on the $^{39}$Ar measurements by performing its calculations in units of moles so that the measurement uncertainty can readily be included. This is accomplished by multiplying the forward-modeled gas fractions ($F_i$) by the total number of moles released during the experiment. Because this value is not directly measured, we add it as a parameter to our model ($\hat{M}_{tot}$) and allow our optimization algorithm to solve for it when using this misfit statistic. We allow the model allowed to choose any $\hat{M}_{tot}$ value within $3\sigma$ of the observed value. Our $\chi^2$ misfit statistic is thereby defined as follows:

$$\chi^2 = \sum_{i=1}^{N} (\frac{M_i - \hat{F}_i \hat{M}_{tot}}{\sigma_i})^2 \tag{4}$$

By contrast, $\%_{frac}$ weights all heating steps equally, regardless of their associated measurement uncertainty. This misfit reports a percent difference between the measured and modeled gas fraction released at each step and is defined as follows:

$$\%_{Frac} = \sum_{i=1}^{N} \frac{F_i - \hat{F}_i}{F_i} \tag{5}$$

where $F_i$ is the measured gas fraction at a given heating step, and $\hat{F}_i$ is the modeled gas fraction at a given heating step.

Regardless of which misfit statistic is used, differential evolution optimizes the parameters in the same way, and begins by generating its initial population. Given a vector of parameters to be tuned for an n-domain model, $\boldsymbol{X}(\hat{M}_{tot}, E_a, \frac{D_0}{a^2}_{(i,i+1,...,n)}, \phi_{(i,i+1,...,n-1)})$, an initial population of candidate vectors is quasi-randomly generated using the Latin Hypercube method (Iman et al., 1981). These vectors ideally capture the full range of the sample space. To avoid user bias, we prescribe search ranges much larger than we imagine realistic for each variable based on prior work (Table 3; Lovera et al., 1997). Lovera et al. (1997).

From here, the improvement process is iterative. To begin, a target vector, $\boldsymbol{X}_i$, which is to be potentially improved, is selected. Next, a multi-step process attempts to replace $\boldsymbol{X}_i$ with an improved offspring vector, $\boldsymbol{X}_i^{'}$, as defined by the misfit function, $g$. First, the best-fit vector, $\boldsymbol{X}_{best}$ (i.e. lowest value of $g(\boldsymbol{X}_i)$), in the current population is copied and modified by adding a scaled value of the difference between two other randomly selected vectors in the population ($\boldsymbol{X}_{r_1}$ and $\boldsymbol{X}_{r_2}$) as

follows:

$$\boldsymbol{U}_i = \boldsymbol{X}_{best} + \beta(\boldsymbol{X}_{r_1} - \boldsymbol{X}_{r_2}) \tag{6}$$

Here, $\beta$ is a uniformly random value between 0.5 and 1.0. Next, $\boldsymbol{U}_i$ is combined with $\boldsymbol{X}_{best}$ to produce the offspring vector $\boldsymbol{X}'_i$. This combination is performed by, for each element of $\boldsymbol{U}_i$, sampling a uniform distribution on [0,1] and replacing the element of $\boldsymbol{X}_i$ with the corresponding element of $\boldsymbol{U}_i$ if the value is less than 0.7, the default recombination constant (see Storn and Price, 1997; Virtanen et al., 2020). In this roundabout way, the trial vector's generation is informed by the existing population.

At this point, $\boldsymbol{X}_i$ has been generated, and if $g(\boldsymbol{X}'_i) < g(\boldsymbol{X}_i)$, $\boldsymbol{X}_i$ is replaced with $\boldsymbol{X}'_i$ in the population. Once the above improvement steps have been performed for every vector in the population, we advance a generation, and population metrics are calculated. Based on the results of these metrics, either another generation is calculated, or the algorithm returns the best-fit vector from the population.

## 5 Synthetic Data Experiments

Synthetic, step-heating diffusion experiments allow for the validation of optimization methods like those described above. For a given set of diffusion kinetics parameters, one can calculate the expected amount of moles released at each step, and these release fractions can then be used as an input to the optimization algorithm to search for the known diffusion kinetics parameters. If the correct parameters are returned, the optimization algorithm is validated. To evaluate the ability of our differential evolution routine (Storn and Price, 1997) in comparison to existing methods (Lovera, 1992; Lovera et al., 1997), we present the results of two such synthetic data experiments (Figure 1).

### 5.1 Synthetic Data Methods

In calculating the number of moles released for each step of a given heating schedule, two assumptions were made. First, $E_a$ was assumed common to all domains (Lovera et al., 1997). Second, we required that the $\ln(\frac{D_0}{a^2})_i$ values of all domains differed by at least 0.25 natural log units. For the purposes of this experiment, any two domains with $\ln(\frac{D_0}{a^2})_i$ values differing by less than 0.25 were considered to be well represented by a model with one fewer domain.

To guide our choices for the synthetic heating schedule and prescribed diffusion kinetics parameters, we used existing literature on K-feldspar diffusion experiments (e.g. Lovera et al., 1997). We begin by defining Experiment A where the heating schedule and $M_{tot}$ were selected from Lovera et al.'s (1997) N13 K-feldspar experiment because this experiment has been published many times (e.g. Lovera et al., 1997; Harrison and Lovera, 2014; Reiners et al., 2017), and because it represents a typical K-feldspar diffusion experiment. $E_a$ was prescribed by randomly sampling from a gaussian distribution with a mean of 192.5 kJ·mol$^{-1}$ and a $\sigma$ of 25, mirroring Lovera et al.'s (1997) database of K-feldspar $E_a$ values. The $\ln(\frac{D_0}{a^2})_i$ values were selected uniformly randomly from a range of 5-25 natural log units. Finally, the $\phi_i$ values were prescribed randomly, requiring

only that the sum of the gas fractions equal 1. The largest of these $\phi_i$ values were intentionally placed in the largest domains,
reflecting common K-feldspar behavior.

Experiment B was then designed to intentionally violate the assumption made by Lovera et al.'s (1997) fitting algorithm
that no domain should significantly degas during early heating steps. This was done by taking the kinetics from experiment A,
removing 1% of the total gas from each domain, and then placing this gas in a new, highly diffusive domain with $\ln(\frac{D}{a^2})_1 = 23.4$, and $\phi_1 = 0.01$.

To calculate the fractional releases and number of moles released after each heating step, we began with the equations for
plane sheet geometry outlined in Ginster and Reiners (2018), which relate each heating step's duration, temperature, and the
domain's $\ln(\frac{D_0}{a^2})_i$ to a fractional release from that domain $((F_{dom})_i)$. To determine the total gas fraction released for a sample
from each heating step $(F_i)$, and not just for a specific domain $((F_{dom})_i)$, we summed the contributions from each domain
as follows: $F_i = \sum_{i=1}^{n}(F_{dom})_i\phi_i$. In finding the number of moles released at each step, we calculated $M_i = F_i \cdot M_{tot}$, where
$M_{tot}$ is the total number of atoms measured in Lovera et al.'s (1997) N13 K-feldspar experiment. To approximate uncertainties,
we simply multiplied the percent error on the $^{40}$Ar/$^{39}$Ar age from each step of the N13 experiment by $M_i$.

Using these synthetic degassing datasets, we attempted to solve for the prescribed diffusion kinetics of both experiments
using our MDD Tool Kit optimization method (Lovera et al., 1997). Because K-feldspar is known to melt above 1100 °C (Luo
et al., 2014), we excluded all calculated heating steps above 1100 °C in our misfit calculations.

## 5.2  Synthetic Data Results and Discussion

The MDD Tool Kit method successfully quantified the diffusion kinetics of both Experiment A and B, returning the correct $E_a$
to within 0.02 kJ·mol$^{-1}$ (Table 4, Figure 1). While the MDD Tool Kit routine did not perform as well in capturing $\phi_{6-8}$ this
is unsurprising since a higher percentage of the gas in those domains was released during the high-temperature steps excluded
from the fitting exercise.

Lovera et al.'s (1997) algorithm correctly quantified the $E_a$ of Experiment A, but it underestimated that of Experiment B
by 9% (Table 4). While we did not implement a Levenberg-Marquardt method (Press, 2007) to solve for each $\ln(\frac{D_0}{a^2})_i$ and $\phi_i$
value, these parameters could not be correct given the incorrectly-predicted $E_a$.

Our synthetic experiment results clearly demonstrate that Lovera et al.'s (1997) algorithm can underestimate $E_a$ (Figure
1). Any sample that contains at least one domain which loses a significant portion of its gas during initial heating steps is
prone to this bias. Importantly, this error is not bi-directional; the Lovera algorithm will systematically underestimate $E_a$. In
contrast, because the optimized $E_a$ does not depend on a linear regression, but instead fits $E_a$ as a free parameter, the MDD
Tool Kit method appears to avoid this bias. This finding suggests that the use of the MDD Tool Kit to reanalyze any MDD-
model-thermochronology study based on the Lovera approach would either produce similar results, or systematically higher
temperatures through geologic time.

## 6 Case Study: Wong et al.'s (2023) Field Validation of the MDD Model

To assess the accuracy of the MDD model, Wong et al. (2023) conducted a field validation study of K-feldspar $^{40}$Ar/$^{39}$Ar thermochronology at the Grayback normal fault block, AZ, USA. This field site is well studied, and several independent thermochronometers have been used here to measure geothermal gradients for application to models of continental extension (Howard and Foster, 1996; Wong et al., 2015).

The field validation primarily relies on four samples from various stratigraphic positions within the Tea Cup pluton, which intruded into the overlying $\sim 1.4$ Ga Ruin Granite at $\sim 70$ Ma (Banks et al., 1972). Samples originally from $\sim 12$ km below the paleo-surface are accessible since these formations were tilted by $\sim 90°$ to the east during mid-Oligocene extension of the Greyback Fault (Banks et al., 1972; Wong et al., 2023). The straightforward stratigraphy and availability of independent thermochronometry data make this location optimal for the validation study.

Wong et al. (2023) found good agreement between their best-fitting $^{40}$Ar/$^{39}$Ar t-T results and existing estimates of the paleo-geothermal gradient prior to mid-Oligocene extension (Howard and Foster, 1996; Wong et al., 2015, 2023). A secondary assessment was provided by fission track, and (U-Th)/He ages in zircon and apatite. Although the observed ages generally do not directly overlap in time with the K-feldspar MDD histories, the results do not preclude one another. In total, their study appears to validate the MDD model. To determine whether the MDD Tool Kit method of fitting diffusion kinetics is consistent with these independent observations, we reanalyze their $^{40}$Ar/$^{39}$Ar stepwise degassing data using our new method.

### 6.1 Methods

#### 6.1.1 Diffusion Kinetics

In determining the diffusion kinetics of each sample, Wong et al. (2023) used a modified version of Lovera et al.'s (1997) fitting algorithm. Their primary modification was to fit an *unweighted* linear regression to the beginning heating steps from each sample to determine $E_a$ and $\ln \frac{D_0}{a_0^2}$. Instead of using the goodness-of-fit metric of Lovera et al. (1997), Wong et al. (2023), varied how many heating steps they included in the regression, and explored the resultant effect of $E_a$ on the constrained t-T path. They then chose the resultant $E_a$ which most closely agreed with independent thermochronological data, noting that the choice of $E_a$ mainly affected the absolute temperature predictions, and not the form of a t-T history. Although this was a routine choice given the tools available at the time of their publication, such user-defined choices may introduce bias, a concern that the MDD Tool Kit inherently circumvents.

In our application of the MDD Tool Kit to Wong et al.'s (2023) diffusion experiment results, we apply both the $\chi^2$ and $\%_{frac}$ misfit statistics defined above. As previously described, heating steps greater than 1100°C were excluded from our misfit calculations.

### 6.1.2 Thermal Paths

When applying K-feldspar MDD modeling, a number of thermal histories are generated using a Monte Carlo-approach to predict an $^{40}$Ar/$^{39}$Ar age spectrum for each prescribed t-T path and using the sample's apparent diffusion kinetics. A misfit is then calculated between the modeled and measured $^{40}$Ar/$^{39}$Ar age spectra to determine the fitness of a particular t-T path.

In our study, we generated 30,000 thermal paths per sample and calculated radiogenic $^{40}$Ar production and diffusive loss for each t-T path using a Crank-Nicholson discretization of the diffusion equation for an infinite sheet geometry (Crank, 1975). We then focus our analysis on the 100 best-fitting t-T paths for each sample based on the following misfit statistic:

$$\chi^2_{Age} = \frac{1}{N} \cdot \sum_{i=1}^{N} \left( \frac{(Age_{Measured})_i - (Age_{Modeled})_i}{(\sigma_{Age_{Measured}})_i} \right)^2 \tag{7}$$

where N is the number of steps included in the thermal modeling, and $\sigma$ is the reported uncertainty on the final $^{40}$Ar/$^{39}$Ar age. As is common practice, Wong et al. (2023) ignored measurements thought to be contaminated with excess $^{40}$Ar ($^{40}$Ar$_E$; Lovera et al., 2002) for comparison with their results, we exclude the same data.

## 6.2 Results

Our predicted $E_a$ values (Tables 5 – 7, Figure 3), regardless of misfit statistic, are systematically higher than those found by Wong et al. (2023) and are at the upper range of those published in Lovera et al.'s (1997) database of K-feldspar diffusion experiments (Figure 1). Given the bias inherent to the regression-fitting method (Figure 1), it is not surprising that our predicted $E_a$ values are systematically higher, since the values of $E_a$ were optimized using the entire dataset, rather than prescribed by linear regression to a subset of user-defined heating steps tuned to agree with independent thermochronological data. And, although we defined a large search space for $E_a$, we find a smaller inter-sample range in predicted $E_a$ than published by Wong et al. (2023), consistent with Lovera et al.'s (1997) observation that the variation of K-feldspar $E_a$ values decreases when more gas is included in the $E_a$ calculation (Tables 5 – 7, Figure 3).

Despite this finding, the choice of misfit statistic appears to influence the calculated diffusion kinetics of a given sample. In our analysis, choosing a $\chi^2$ misfit instead of the $\%_{frac}$ misfit statistic led to intra-sample differences in the predicted $E_a$ between 4.4 – 22.4 kJ·mol$^{-1}$ (Tables 8 and 5, respectively). For example, there is a 22.4 kJ·mol$^{-1}$ difference between the best-fit $E_a$ value for sample GR-2 when using the $\chi^2$ misfit compared to the $\%_{frac}$ misfit (Table 6). This disparity may provide an estimate of the uncertainty on the $E_a$, with the true $E_a$ lying anywhere between those values. We recommend that investigators using the MDD_toolkit consider model fits generated by both the $\chi^2$ and the $\%_{frac}$ misfit statistics as equally plausible, given the lack of justification for choosing one over the other.

An additional noteworthy model behavior emerged during our analysis. In three cases, the MDD Tool Kit identified at least one retentive domain that exhibited no gas diffusion throughout the simulated experiment, as indicated in Tables 4 – 7. The total $\phi$ value(s) of these domains typically equaled the gas quantity contained in the heating steps above 1100 °C. While the result that no gas diffused from the most retentive domain (or domains) during the simulated experiment may appear to suggest a poor

model choice, this is not necessarily the case. Since K-feldspar begins to melt above 1100 °C, the MDD model is simply not applicable above this temperature. Furthermore, any domain retaining all its gas during simulation will yield identical release fractions, thereby making the model's fit insensitive to retentivities above a specific threshold (assuming all other parameters remain constant). Given our deliberate inclusion of a wide search space for each diffusion parameter and the stochastic nature of our optimization algorithm, such behavior is to be expected.

### 6.3 Reinterpretation of Field Calibration

Although our reinterpretation of these data largely finds t-T paths of similar form to those of Wong et al. (2023), our absolute temperature predictions are about $60 – 75$ °C higher (Figure 4). To demonstrate that our approach yields results that are equally permissible, we compare them to the independent t-T constraints that exist at the Grayback fault.

Wong et al. (2023) proposed three criteria to assess the validity of an MDD model: (i) The thermal histories should be consistent with the stratigraphic heights of the samples, (ii) The form of the predictions should match prior work, including the timing, rate, and duration of cooling events, and (iii) The absolute temperatures should agree with those predicted by previous estimates of the pre-extensional geothermal gradient (Howard and Foster, 1996; Wong et al., 2015). The constrained t-T paths shown in Wong et al., (2023) and our re-analysis of the same dataset using the MDD Tool Kit both meet these criteria (Figure 4).

The predicted t-T paths for all four samples generally align with their respective stratigraphic positions; samples from higher stratigraphic heights reflect the oldest portions of the thermal histories (Figure 4). Our findings reveal that the sample situated at the highest stratigraphic level (GR-1) cooled below its Ar closure temperature around $\sim 55$ Ma, indicating relatively rapid cooling between $\sim 70 – 55$ Ma. This trend is consistent with Howard and Foster's (1996) interpretation that the most shallow depths of the pluton experienced rapid cooling as equilibrated with ambient temperatures.

MDD results for the next sample (GR-2), slightly deeper than GR-1, suggest a gradual cooling rate of approximately $\sim 5$ °C/Ma between $\sim 55 – 30$ Ma, consistent with the prediction of $\sim 4 – 6$ °C/Ma by Howard and Foster (1996).

The two samples at the lowest stratigraphic levels, GR-27 and GR-8, exhibit similar temperature histories. The median pathway for GR-27 calculated with the $\chi^2$ misfit indicates slightly higher temperatures compared to GR-8, despite the latter occupying a slightly lower stratigraphic depth. However, given that their thermal pathways are within $1\sigma$ of each other, it suggests that this technique may not be capable of resolving subtle differences between these samples. Moreover, the presence of significant $Ar_E$ (Lovera et al., 2002) in sample GR-8 introduces additional uncertainty into its Paleogene thermal history. Despite the presence of $Ar_E$ (Lovera et al., 2002), both of these deep stratigraphic samples demonstrate rapid cooling commencing at around $\sim 27$-28 Ma, consistent with the predicted timing of this cooling based on apatite fission track (AFT) ages (Howard and Foster, 1996).

Finally, all four of our newly-calculated MDD models predict similar absolute temperatures to those estimated of the paleo-geothermal gradient along the Tea Cup pluton prior to onset of extension along the Grayback fault. The paleo-gradient has been estimated from temperature predictions made at 27 Ma at a variety of paleodepths using AFT (Howard and Foster, 1996). While our estimates are consistently warmer than those calculated by Wong et al. (2023), they are still within the uncertainty

bounds of the predicted thermal gradient (Figure 5). The overall agreement between the rates and timing of cooling, as well as the relationship between relative temperatures over time and the samples' stratigraphic order, indicates that the MDD Tool Kit generates t-T results that are not excluded by the existing geologic and independent thermochronology data.

## 7 Conclusions

The Multiple Diffusion Domain model for $^{40}$Ar/$^{39}$Ar thermochronology is valuable because it allows one to constrain a mineral's continuous thermal history through geologic time (McDougall and Harrison, 1999). However, the methods previously used to empirically fit stepwise degassing data have required unverifiable assumptions about the grain size distribution of a sample. Further, the commonly used method of quantifying $E_a$ by fitting an uncertainty-weighted linear regression to the lowest-temperature degassing steps of an experiment does not reliably return the correct values, but instead inadvertently introduces a user bias. When this method fails to predict the correct value, it underestimates the true value. To address these limitations, we present a new numerical routine that does not require fitting a linear regression to a user-defined subset of heating steps to quantify a sample's $E_a$. Our new method utilizes a differential evolution algorithm to robustly search the MDD-parameter space and solve for all parameters simultaneously. The code, entitled MDD Tool Kit, is open-source, pip-installable, and available on GitHub. To evaluate its validity, we apply this new method to reinterpret the dataset published in Wong et al.'s (2023) field validation of the $^{40}$Ar/$^{39}$Ar K-feldspar MDD system at the Grayback fault. The diffusion kinetics fit by the MDD Tool Kit predict t-T paths that are consistent with independent observations and geologic constraints, and on average, 60 – 75 °C warmer than those previously published. We attribute this temperature difference to biases potentially introduced by the previously used modeling strategies.

*Code availability.* The code released in this publication is available at our github link: https://github.com/dgorin1/mddtoolkit

## Appendix A: Supplemental Tables

For completeness, we include the optimization inputs used to reanalyze the experiments performed by Wong et al. (2023) (Tables A1–A4).

*Author contributions.* ALG and DLS conceived of the study. ALG and JMG designed the software. ALG, DLS, and MB contributed to the theory behind the software. ALG and DLS led manuscript writing. ALG constructed figures. All authors reviewed and edited the manuscript.

*Competing interests.* Authors declare that they have no competing interests.

*Acknowledgements.* We acknowledge funding from the Ann and Gordon Getty foundation. We also acknowledge Oscar Lovera for his

insightful discussions about the theory of the MDD model. Finally, we thank Martin Wong for providing us with their data and for calculating

measured $^{39}$Ar uncertainties for their data.

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

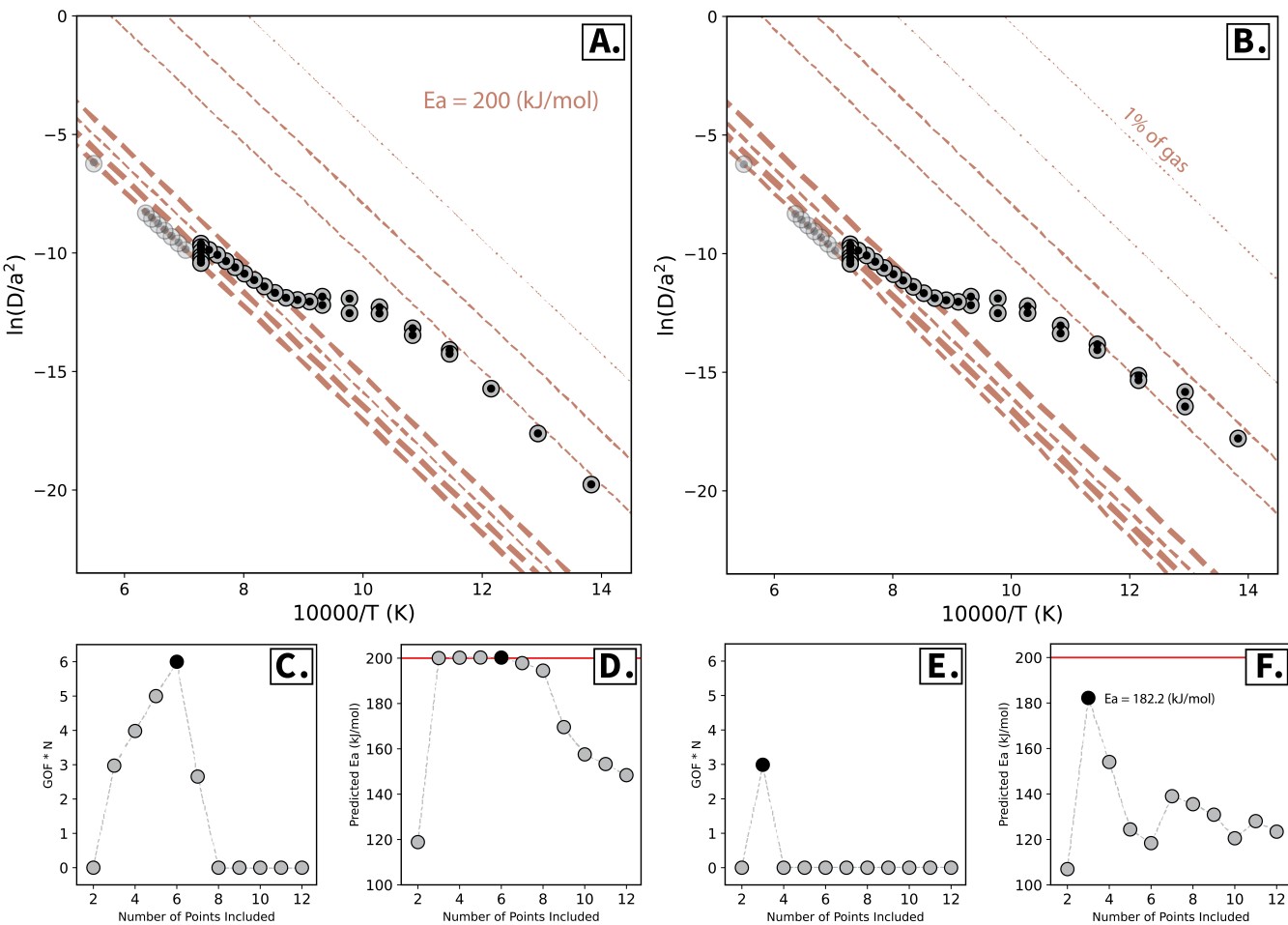

**Figure 1. A.** Arrhenius plot resulting from synthetic Experiment A. Gray Circles show the expected $\frac{D}{a^2}$ values calculated from the pre-determined diffusion kinetics (Table 4), and transparent gray circles show the heating steps excluded from the fitting exercise. Black circles represent the values predicted by the MDD Tool Kit method. **B.** Same as A. except showing the expected $\frac{D}{a^2}$ values calculated from the pre-determined diffusion kinetics for Experiment B. **C.** Value of $q \cdot N$ as the number of points included in the uncertainty-weighted linear regression increases for Experiment A. Lovera et al.'s (1997) algorithm for determining the $E_a$ maximizes the value of this function to determine the appropriate number of points to include. The black dot indicates this value. **D.** The predicted $E_a$ as one increases the number of points included in the unweighted linear regression. The black point indicates the $E_a$ selected by Lovera et al.'s (1997) algorithm for Experiment A. In this case, the algorithm selects the correct $E_a$. E. Same as panel C, but showing the results for synthetic Experiment B. **F.** Same as panel D but for synthetic Experiment B. In this case, Lovera et al.'s (1997) algorithm underestimates the correct $E_a$.

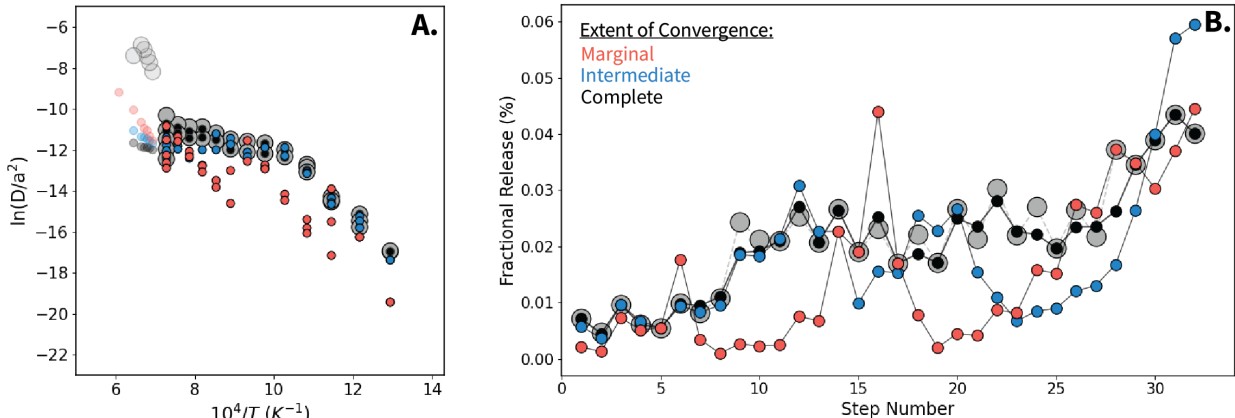

**Figure 2. A.** Results of an MDD Tool Kit optimization applied to Wong et al.'s (2023) sample GR-27 demonstrating the iterative nature of the improvement using the $\%_{frac}$ misfit statistic shown in Arrhenius space. **B.** Same as A, but shown in the space where the optimization is performed–the fractional release at each heating step. Red circles show the optimization halted after 8 iterations ($\%_{frac} = 15.06$), blue circles show the same optimization halted after 28 iterations ($\%_{frac} = 8.33$), and black circles show the same optimization results after complete convergence after 18,419 iterations ($\%_{frac} = 1.82$). The number of iterations is independently determined by the differential evolution algorithm using the convergence criteria outlined by Virtanen et al. (2020). The gray circles show the observed results.

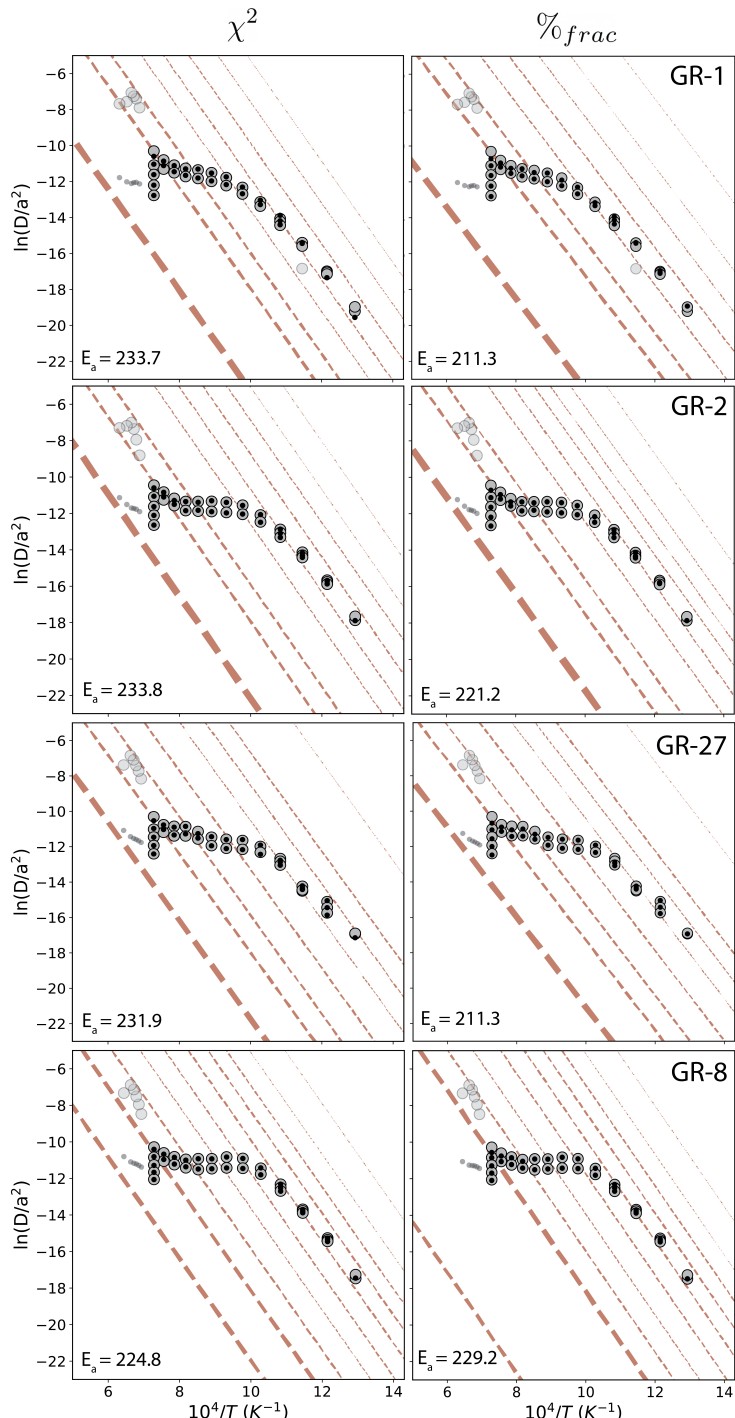

**Figure 3.** Arrhenius plots showing our reanalysis of Wong et al.'s (2023) samples with both the $\chi^2$ and $\%_{frac}$ misfit statistics. Red lines represent the diffusion kinetics of each individual domain, and their thicknesses are proportional to domain's $\phi$ value. Gray circles show the experimental results and black dots show the MDD model predictions. All $E_a$ are given in kJ·mol$^{-1}$

## Predicted Thermal History          ## ⁴⁰Ar/³⁹Ar Age Spectrum

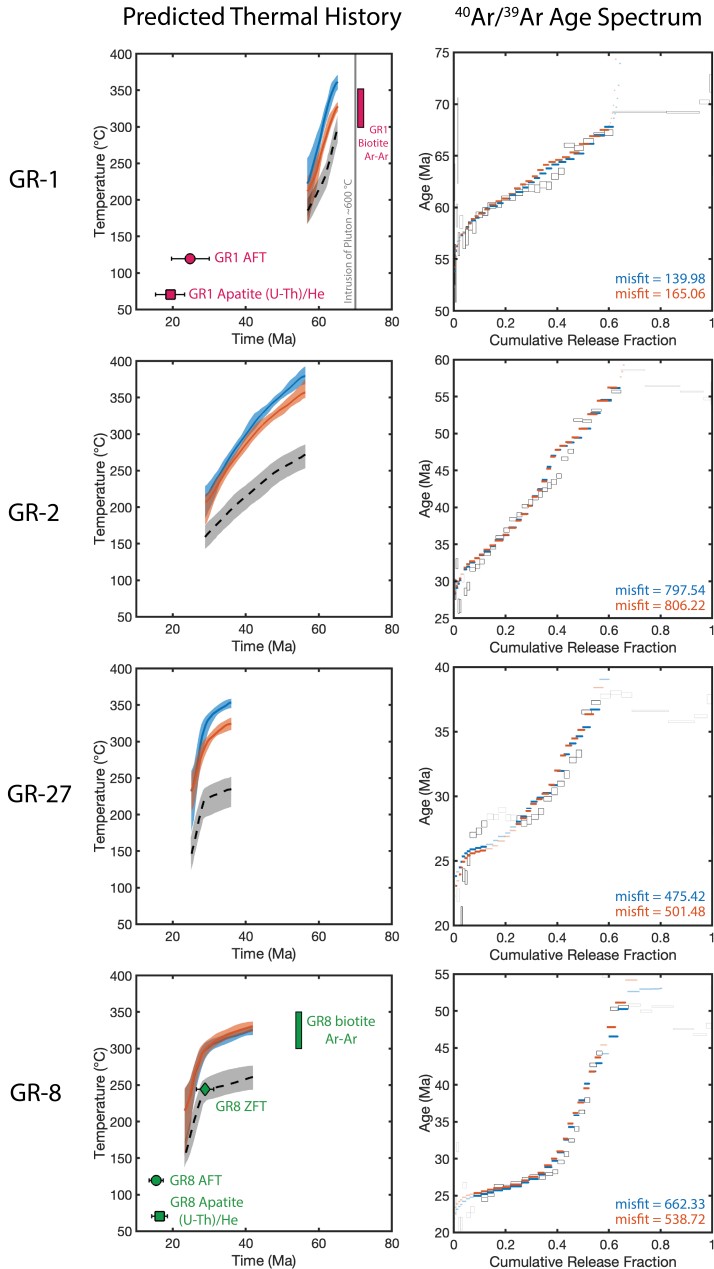

**Figure 4.** Resulting best-fit thermal pathways and ⁴⁰Ar-³⁹Ar age spectra from our reanalysis of Wong et al.'s (2023) field validation of the MDD method in K-feldspar. Blue shaded regions in the thermal history plots correspond to the $\chi^2$ misfit statistic. The orange shaded regions correspond to the $\%_{frac}$ misfit statistic. More specifically, the shaded regions represent a $1\sigma$ deviation from the median of the top 100 best-fitting thermal paths for a given sample (plotted in bold). The dotted line and gray regions represent Wong et al.'s (2023) predictions for the same samples. The ⁴⁰Ar/³⁹Ar age spectrum plots show the predicted ⁴⁰Ar/³⁹Ar Ages for the median $\chi^2$ (blue) and $\%_{frac}$ (orange) thermal paths. Grey boxes represent the measured values from Wong (2023). Heating steps excluded from the modeling exercise are made to be transparent.

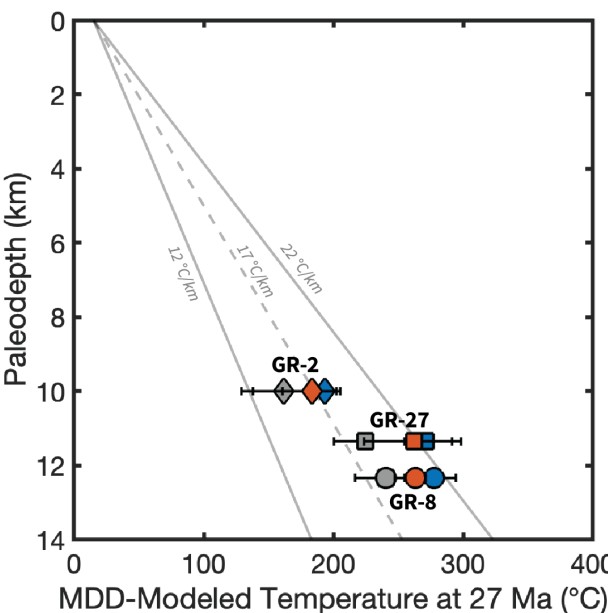

**Figure 5.** Predicted paleotemperatures at 27 Ma from samples GR-2, GR-27, and GR-8. The gray symbols represent Wong et al.'s (2023) results, the orange symbols represent our results from the $\%_{frac}$ misfit statistic, and the blue symbols represent our results from the $\chi^2$ misfit statistic.

**Table 1.** Table of variables.

| Variable | Description | Units |
|:---:|:---:|:---:|
| MDD Model | | |
| $D$ | Diffusivity | $cm^2 \cdot s^1$ |
| $D_0$ | Diffusivity at infinite temperature | $cm^2 \cdot s^1$ |
| $E_a$ | Activation energy | $kJ \cdot mol^{-1}$ |
| $R$ | Gas constant | $kJ \cdot mol^{-1} \cdot K^{-1}$ |
| $T$ | Absolute temperature | K |
| $a$ | Radius of diffusion domain | cm |
| $\phi_i$ | Proportion of total gas contained in domain $i$ | – |
| Lovera et al. (1997) Optimization Method | | |
| $n$ | Number of domains in an MDD model | – |
| $q$ | goodness-of-fit probability used by Lovera et al. (1997) | – |
| $\chi^2_{lovera}$ | Misfit statistic used by Lovera et al. (1997) | – |
| $\left(\frac{D}{a^2}\right)_i$ | Observed pre-exponential term for heating step $i$ | $s^{-1}$ |
| $\left(\frac{\hat{D}}{a^2}\right)_i$ | Modeled pre-exponential term for heating step $i$ | $s^{-1}$ |
| $\sigma_i$ | Observed pre-exponential 1 S.D. for heating step $i$ | $s^{-1}$ |
| $N$ | Number of heating steps included in optimization | – |
| $a_0$ | Grain size of reference domain (Lovera et al., 1997) | cm |
| MDD_toolkit | | |
| $\chi^2$ | Misfit statistic used in MDD_toolkit defined in equation 4 | – |
| $\%_{frac}$ | Misfit statistic used in MDD_toolkit defined in equation 5 | – |
| $M_i$ | Measured $^{39}$Ar released at heating step $i$ | mol |
| $\hat{M}_i$ | Modeled $^{39}$Ar released at heating step $i$ | mol |
| $M_{tot}$ | Measured total $^{39}$Ar released during experiment | mol |
| $\hat{M}_{tot}$ | Modeled total $^{39}$Ar released during experiment | mol |
| $F_i$ | Measured fraction of $M_{tot}$ released at heating step $i$ | – |
| $\hat{F}_i$ | Modeled fraction of $M_{tot}$ released at heating step $i$ | – |
| Differential Evolution | | |
| $\boldsymbol{X}$ | Vector of MDD-model parameters | – |
| $\boldsymbol{X'}$ | Offspring vector of $\boldsymbol{X}$ | – |
| $g$ | Generic misfit function | – |
| $\boldsymbol{X_{best}}$ | Vector in population with lowest value of $g(\boldsymbol{X}_i)$ | – |
| $\boldsymbol{X_r}$ | Random vector selected from population | – |
| $\beta$ | Value between 0.5 and 1.0 used to scale difference between two $\boldsymbol{X_r}$ | – |
| Continued on next page... | | |

| Variable | Description | Units |
|---|---|---|
| $\boldsymbol{U_i}$ | Intermediate vector defined in Equation 6 | – |
| Thermal Paths | | |
| $\chi^2_{age}$ | Misfit statistic used to determine best-fitting thermal path defined in equation 7 | – |
| $(Age_{Measured})_i$ | Measured $^{40}Ar/^{39}Ar$ | Ma |
| $(Age_{Modeled})_i$ | Modeled $^{40}Ar/^{39}Ar$ | Ma |
| $(\sigma_{Age_{Measured}})_i$ | Observed age S.D. for heating step $i$ | Ma |

**Table 2.** Equations used by the MDD Tool Kit to calculate $f$, the fractional loss from each domain during each heating step in a stepwise degassing experiment (Crank, 1975; Ginster and Reiners, 2018).

| Geometry | Equation | Validity |
|---|---|---|
| Sphere (radius $a$) | $f = 1 - \frac{6}{\pi^2} \sum_1^\infty \frac{1}{n^2} \exp(\frac{-n^2\pi^2 Dt}{a^2})$ | All $f$ |
| | $f \approx 1 - \frac{6}{\pi^2} \exp(\frac{-\pi^2 Dt}{a^2})$ | $0.85 \leq f \leq= 1$ |
| | $f \approx \frac{6}{\pi^{\frac{3}{2}}} \sqrt{\frac{\pi^2 Dt}{a^2}} - \frac{3Dt}{a^2}$ | $0 \leq f \leq 0.85$ |
| Plane Sheet (half-width $a$) | $f = 1 - \frac{8}{\pi^2} \sum_1^\infty \frac{1}{(2n+1)^2} \exp[-(2n+1)^2 \pi^2 \frac{Dt}{4a^2}]$ | All $f$ |
| | $f \approx 1 - \frac{8}{\pi^2} \exp(\frac{\pi^2 Dt}{4a^2})$ | $0.45 \leq f \leq 1$ |
| | $f \approx \frac{2}{\sqrt{\pi}} \sqrt{\frac{Dt}{a^2}}$ | $0 \leq f \leq 0.60$ |

**Table 3.** Parameter search ranges used in this publication. These values can be adjusted in the open-access code.

| Parameter | Search Range |
|---|---|
| $\hat{M}_{tot}$ (mol) | $(\hat{M}_{tot} - 3\sigma, \hat{M}_{tot} + 3\sigma)$ |
| $E_a$ (kJ/mol) | (50, 500) |
| $\ln\left(\frac{D_0}{a^2}\right)$ $(s^{-1})$ | (-5, 50) |
| $\phi$ (normalized units) | (0, 1) |

**Table 4.** Setup and results of synthetic diffusion experiment optimizations. Experiment A and B are the same except that a domain with $\ln(\frac{D_0}{a^2})_1$ has been added to experiment B. To redistribute the gas, 1% was removed from all other domains such that $\phi_1 = 0.01$.

| | Experiment A | Optimization Results A | Experiment B | Optimization Results B |
|---|---|---|---|---|
| $M_{tot}$ ($\cdot 10^{-12}$) | 5.21 | 5.21 | 5.21 | 5.21 |
| $E_a$ ($\frac{kJ}{mol}$) | 200.2 | 200.0 | 200.2 | 200.3 |
| $\ln(\frac{D_0}{a^2})_1$ | 19.45 | 19.42 | 23.8 | 23.81 |
| $\ln(\frac{D_0}{a^2})_2$ | 16.18 | 16.14 | 19.45 | 19.46 |
| $\ln(\frac{D_0}{a^2})_3$ | 13.93 | 13.90 | 16.18 | 16.19 |
| $\ln(\frac{D_0}{a^2})_4$ | 8.89 | 8.88 | 13.93 | 13.94 |
| $\ln(\frac{D_0}{a^2})_5$ | 8.09 | 8.18 | 8.89 | 8.89 |
| $\ln(\frac{D_0}{a^2})_6$ | 7.51 | 7.58 | 8.09 | 8.04 |
| $\ln(\frac{D_0}{a^2})_7$ | 6.98 | 7.02 | 7.51 | 7.49 |
| $\ln(\frac{D_0}{a^2})_8$ | – | – | 6.98 | 6.97 |
| $\phi_1$ | 0.025 | 0.025 | 0.01 | 0.01 |
| $\phi_2$ | 0.097 | 0.097 | 0.025 | 0.025 |
| $\phi_3$ | 0.082 | 0.082 | 0.096 | 0.096 |
| $\phi_4$ | 0.226 | 0.214 | 0.082 | 0.082 |
| $\phi_5$ | 0.127 | 0.109 | 0.224 | 0.233 |
| $\phi_6$ | 0.273 | 0.252 | 0.125 | 0.129 |
| $\phi_7$ | 0.170 | 0.220 | 0.270 | 0.266 |
| $\phi_8$ | – | – | 0.168 | 0.160 |

**Table 5.** Results from sample GR-1.

| Sample GR-1 Diffusion Kinetics | Wong (2023) Results | $\chi^2$ (This Work) | $\%_{frac}$ (This Work) |
|---|---|---|---|
| $E_a\ (\frac{kJ}{mol})$ | 187.1 | 233.7 | 211.3 |
| $\ln(\frac{D_0}{a^2})_1$ | 14.63 | 22.8 | 23.1 |
| $\ln(\frac{D_0}{a^2})_2$ | 12.57 | 20.5 | 19.3 |
| $\ln(\frac{D_0}{a^2})_3$ | 10.75 | 18.3 | 17.0 |
| $\ln(\frac{D_0}{a^2})_4$ | 7.85 | 16.3 | 14.8 |
| $\ln(\frac{D_0}{a^2})_5$ | 7.76 | 14.50 | 12.8 |
| $\ln(\frac{D_0}{a^2})_6$ | 6.05 | 12.1 | 10.1 |
| $\ln(\frac{D_0}{a^2})_7$ | 3.77 | 10.2 | 8.2 |
| $\ln(\frac{D_0}{a^2})_8$ | 1.14 | 4.4 | 1.9 |
| $\phi_1$ | 0.049 | 0.022 | 0.001 |
| $\phi_2$ | 0.088 | 0.044 | 0.026 |
| $\phi_3$ | 0.094 | 0.073 | 0.056 |
| $\phi_4$ | 0.111 | 0.080 | 0.090 |
| $\phi_5$ | 0.096 | 0.058 | 0.085 |
| $\phi_6$ | 0.127 | 0.145 | 0.148 |
| $\phi_7$ | 0.101 | 0.187 | 0.196 |
| $\phi_8$ | 0.033 | 0.391 | 0.398 |

**Table 6.** Results from sample GR-2.

| Sample GR-2 Diffusion Kinetics | Wong (2023) Results | $\chi^2$ (This Work) | $\%_{frac}$ |
|---|---|---|---|
| $E_a \left(\frac{kJ}{mol}\right)$ | 182.0 | 233.8 | 221.2 |
| $\ln(\frac{D_0}{a^2})_1$ | 17.36 | 25.7 | 23.2 |
| $\ln(\frac{D_0}{a^2})_2$ | 15.43 | 23.0 | 20.5 |
| $\ln(\frac{D_0}{a^2})_3$ | 13.65 | 21.2 | 18.9 |
| $\ln(\frac{D_0}{a^2})_4$ | 11.76 | 19.4 | 17.7 |
| $\ln(\frac{D_0}{a^2})_5$ | 8.49 | 17.4 | 16.0 |
| $\ln(\frac{D_0}{a^2})_6$ | 8.46 | 15.5 | 14.2 |
| $\ln(\frac{D_0}{a^2})_6$ | 6.53 | 12.2 | 11.0 |
| $\ln(\frac{D_0}{a^2})_8$ | 4.44 | 10.2 | 9.1 |
| $\ln(\frac{D_0}{a^2})_9$ | 1.79 | 5.9 | 4.9 |
| $\phi_1$ | 0.036 | 0.009 | 0.011 |
| $\phi_2$ | 0.092 | 0.022 | 0.045 |
| $\phi_3$ | 0.096 | 0.066 | 0.052 |
| $\phi_4$ | 0.074 | 0.081 | 0.067 |
| $\phi_5$ | 0.05 | 0.074 | 0.072 |
| $\phi_6$ | 0.119 | 0.064 | 0.069 |
| $\phi_7$ | 0.145 | 0.134 | 0.121 |
| $\phi_8$ | 0.076 | 0.157 | 0.154 |
| $\phi_9$ | 0.312 | 0.393 | 0.410 |

**Table 7.** Results from sample GR-27.

| Sample GR-27 Diffusion Kinetics | Wong (2023) Results | $\chi^2$ (This Work) | Percent_Frac (This Work) |
|---|---|---|---|
| $E_a$ $(\frac{kJ}{mol})$ | 159.5 | 231.9 | 211.3 |
| $\ln(\frac{D_0}{a^2})_1$ | 11.2 | 26.3 | 23.3 |
| $\ln(\frac{D_0}{a^2})_2$ | 9.9 | 21.2 | 18.2 |
| $\ln(\frac{D_0}{a^2})_3$ | 7.5 | 19.4 | 16.7 |
| $\ln(\frac{D_0}{a^2})_4$ | 5.2 | 17.4 | 15.0 |
| $\ln(\frac{D_0}{a^2})_5$ | 5.0 | 14.6 | 12.6 |
| $\ln(\frac{D_0}{a^2})_6$ | 2.9 | 12.0 | 10.1 |
| $\ln(\frac{D_0}{a^2})_6$ | 2.0 | 9.9 | 8.1 |
| $\ln(\frac{D_0}{a^2})_8$ | -0.1 | 6.1 | 4.3 |
| $\phi_1$ | 0.124 | 0.015 | 0.013 |
| $\phi_2$ | 0.064 | 0.092 | 0.098 |
| $\phi_3$ | 0.109 | 0.077 | 0.070 |
| $\phi_4$ | 0.117 | 0.053 | 0.050 |
| $\phi_5$ | 0.102 | 0.114 | 0.103 |
| $\phi_6$ | 0.104 | 0.130 | 0.121 |
| $\phi_7$ | 0.112 | 0.171 | 0.168 |
| $\phi_8$ | 0.268 | 0.348 | 0.377 |

**Table 8.** Results from sample GR-8.

| Sample GR-8 Diffusion Kinetics | Wong (2023) Results | $\chi^2$ (This Work) | $\%_{frac}$ |
|---|---|---|---|
| $E_a\ (\frac{kJ}{mol})$ | 182.0 | 224.8 | 229.2 |
| $\ln(\frac{D_0}{a^2})_1$ | 14.9 | 24.2 | 25.0 |
| $\ln(\frac{D_0}{a^2})_2$ | 13.5 | 21.1 | 21.9 |
| $\ln(\frac{D_0}{a^2})_3$ | 11.9 | 19.3 | 19.8 |
| $\ln(\frac{D_0}{a^2})_4$ | 10.2 | 17.5 | 17.8 |
| $\ln(\frac{D_0}{a^2})_5$ | 7.5 | 15.6 | 15.9 |
| $\ln(\frac{D_0}{a^2})_6$ | 7.5 | 13.7 | 13.6 |
| $\ln(\frac{D_0}{a^2})_6$ | 5.5 | 11.4 | 11.8 |
| $\ln(\frac{D_0}{a^2})_8$ | 4.3 | 9.0 | 10.0 |
| $\ln(\frac{D_0}{a^2})_9$ | 2.7 | -4.1 | 7.5 |
| $\phi_1$ | 0.053 | 0.009 | 0.008 |
| $\phi_2$ | 0.152 | 0.048 | 0.050 |
| $\phi_3$ | 0.111 | 0.125 | 0.136 |
| $\phi_4$ | 0.080 | 0.109 | 0.106 |
| $\phi_5$ | 0.075 | 0.082 | 0.085 |
| $\phi_6$ | 0.074 | 0.058 | 0.062 |
| $\phi_7$ | 0.143 | 0.118 | 0.081 |
| $\phi_8$ | 0.061 | 0.260 | 0.107 |
| $\phi_9$ | 0.252 | 0.191 | 0.365 |

**Table A1.** Sample GR-1 input (Wong et al., 2023).

| Step Number | Temperature (°C) | Duration (Hours) | $^{39}$Ar Released (mol) | Measurement Uncertainty (mol; $1\sigma$) | Included in Diffusion Kinetics Optimization? (1=yes, 0=no) |
|---|---|---|---|---|---|
| 0 | 500 | 0.250 | 2.54E-16 | 8.40E-19 | 1 |
| 1 | 500 | 0.417 | 1.95E-16 | 7.58E-19 | 1 |
| 2 | 550 | 0.250 | 4.44E-16 | 1.22E-18 | 1 |
| 3 | 550 | 0.333 | 3.50E-16 | 1.05E-18 | 1 |
| 4 | 550 | 0.500 | 3.58E-16 | 8.48E-19 | 1 |
| 5 | 600 | 0.250 | 7.40E-16 | 1.29E-18 | 1 |
| 6 | 600 | 0.333 | 1.89E-16 | 7.23E-19 | 0 |
| 7 | 600 | 0.500 | 8.21E-16 | 1.57E-18 | 1 |
| 8 | 650 | 0.250 | 1.37E-15 | 1.87E-18 | 1 |
| 9 | 650 | 0.333 | 1.19E-15 | 1.59E-18 | 1 |
| 10 | 650 | 0.500 | 1.18E-15 | 1.43E-18 | 1 |
| 11 | 700 | 0.250 | 1.77E-15 | 2.23E-18 | 1 |
| 12 | 700 | 0.333 | 1.54E-15 | 1.75E-18 | 1 |
| 13 | 750 | 0.250 | 2.68E-15 | 2.59E-18 | 1 |
| 14 | 750 | 0.333 | 2.06E-15 | 2.29E-18 | 1 |
| 15 | 800 | 0.250 | 3.24E-15 | 2.84E-18 | 1 |
| 16 | 800 | 0.333 | 2.37E-15 | 2.58E-18 | 1 |
| 17 | 850 | 0.250 | 3.19E-15 | 2.79E-18 | 1 |
| 18 | 850 | 0.333 | 2.34E-15 | 2.40E-18 | 1 |
| 19 | 900 | 0.250 | 2.99E-15 | 3.27E-18 | 1 |
| 20 | 900 | 0.333 | 2.27E-15 | 2.22E-18 | 1 |
| 21 | 950 | 0.250 | 2.76E-15 | 2.62E-18 | 1 |
| 22 | 950 | 0.333 | 2.17E-15 | 2.14E-18 | 1 |

| Step Number | Temperature (°C) | Duration (Hours) | $^{39}$Ar Released (mol) | Measurement Uncertainty (mol; $1\sigma$) | Included in Diffusion Kinetics Optimization? (1=yes, 0=no) |
|---|---|---|---|---|---|
| 23 | 1000 | 0.250 | 2.86E-15 | 2.47E-18 | 1 |
| 24 | 1000 | 0.333 | 2.34E-15 | 2.47E-18 | 1 |
| 25 | 1050 | 0.250 | 3.34E-15 | 3.08E-18 | 1 |
| 26 | 1050 | 0.333 | 2.51E-15 | 2.30E-18 | 1 |
| 27 | 1100 | 0.250 | 4.76E-15 | 3.70E-18 | 1 |
| 28 | 1100 | 0.500 | 4.01E-15 | 3.26E-18 | 1 |
| 29 | 1100 | 1.000 | 4.41E-15 | 2.98E-18 | 1 |
| 30 | 1100 | 2.000 | 4.54E-15 | 3.74E-18 | 1 |
| 31 | 1100 | 3.333 | 3.73E-15 | 7.08E-18 | 1 |
| 32 | 1180 | 0.233 | 2.29E-14 | 8.27E-18 | 0 |
| 33 | 1200 | 0.233 | 1.41E-14 | 7.50E-18 | 0 |
| 34 | 1215 | 0.233 | 4.24E-15 | 3.30E-18 | 0 |
| 35 | 1230 | 0.233 | 1.07E-15 | 1.46E-18 | 0 |
| 36 | 1260 | 0.233 | 1.44E-16 | 6.39E-19 | 0 |
| 37 | 1310 | 0.233 | 4.61E-17 | 4.30E-19 | 0 |
| 38 | 1370 | 0.233 | 2.87E-17 | 3.34E-19 | 0 |

**Table A2.** Sample GR-2 input (Wong et al., 2023).

| Step Number | Temperature (°C) | Duration (Hours) | $^{39}$Ar Released (mol) | Measurement Uncertainty (mol; $1\sigma$) | Included in Diffusion Kinetics Optimization? (1=yes, 0=no) |
|---|---|---|---|---|---|
| 0 | 500 | 0.250 | 6.43E-16 | 1.22E-18 | 1 |
| 1 | 500 | 0.417 | 4.82E-16 | 1.09E-18 | 1 |
| 2 | 550 | 0.250 | 1.11E-15 | 1.62E-18 | 1 |
| 3 | 550 | 0.333 | 8.51E-16 | 1.41E-18 | 1 |
| 4 | 550 | 0.500 | 8.55E-16 | 1.47E-18 | 1 |
| 5 | 600 | 0.250 | 1.79E-15 | 2.02E-18 | 1 |
| 6 | 600 | 0.333 | 1.51E-15 | 1.79E-18 | 1 |
| 7 | 600 | 0.500 | 1.62E-15 | 2.12E-18 | 1 |
| 8 | 650 | 0.250 | 2.94E-15 | 2.47E-18 | 1 |
| 9 | 650 | 0.333 | 2.59E-15 | 2.83E-18 | 1 |
| 10 | 650 | 0.500 | 2.54E-15 | 2.81E-18 | 1 |
| 11 | 700 | 0.250 | 3.58E-15 | 2.98E-18 | 1 |
| 12 | 700 | 0.333 | 2.80E-15 | 2.49E-18 | 1 |
| 13 | 750 | 0.250 | 4.49E-15 | 3.20E-18 | 1 |
| 14 | 750 | 0.333 | 3.17E-15 | 2.08E-18 | 1 |
| 15 | 800 | 0.250 | 4.10E-15 | 2.59E-18 | 1 |
| 16 | 800 | 0.333 | 2.83E-15 | 2.30E-18 | 1 |
| 17 | 850 | 0.250 | 3.57E-15 | 3.04E-18 | 1 |
| 18 | 850 | 0.333 | 2.54E-15 | 2.42E-18 | 1 |
| 19 | 900 | 0.250 | 3.09E-15 | 2.67E-18 | 1 |
| 20 | 900 | 0.333 | 2.30E-15 | 2.23E-18 | 1 |
| 21 | 950 | 0.250 | 2.77E-15 | 2.38E-18 | 1 |
| 22 | 950 | 0.333 | 2.15E-15 | 2.61E-18 | 1 |

495

| Step Number | Temperature (°C) | Duration (Hours) | $^{39}$Ar Released (mol) | Measurement Uncertainty (mol; $1\sigma$) | Included in Diffusion Kinetics Optimization? (1=yes, 0=no) |
|---|---|---|---|---|---|
| 23 | 1000 | 0.250 | 2.99E-15 | 2.92E-18 | 1 |
| 24 | 1000 | 0.333 | 2.57E-15 | 2.73E-18 | 1 |
| 25 | 1050 | 0.250 | 3.82E-15 | 3.57E-18 | 1 |
| 26 | 1050 | 0.333 | 3.15E-15 | 3.13E-18 | 1 |
| 27 | 1100 | 0.250 | 4.97E-15 | 3.45E-18 | 1 |
| 28 | 1100 | 0.500 | 4.89E-15 | 3.55E-18 | 1 |
| 29 | 1100 | 1.000 | 5.39E-15 | 3.80E-18 | 1 |
| 30 | 1100 | 2.000 | 5.82E-15 | 3.99E-18 | 1 |
| 31 | 1100 | 3.333 | 5.15E-15 | 3.53E-18 | 1 |
| 32 | 1180 | 0.233 | 1.36E-14 | 5.42E-18 | 0 |
| 33 | 1200 | 0.233 | 1.95E-14 | 6.50E-18 | 0 |
| 34 | 1215 | 0.233 | 1.32E-14 | 5.65E-18 | 0 |
| 35 | 1230 | 0.233 | 4.09E-15 | 3.01E-18 | 0 |
| 36 | 1260 | 0.233 | 5.86E-16 | 1.14E-18 | 0 |
| 37 | 1310 | 0.233 | 1.16E-16 | 5.24E-19 | 0 |
| 38 | 1370 | 0.233 | 3.83E-17 | 2.94E-19 | 0 |

**Table A3.** Sample GR-8 input (Wong et al., 2023).

| Step Number | Temperature (°C) | Duration (Hours) | $^{39}$Ar Released (mol) | Measurement Uncertainty (mol; $1\sigma$) | Included in Diffusion Kinetics Optimization? (1=yes, 0=no) |
|---|---|---|---|---|---|
| 0 | 500 | 0.250 | 8.31E-16 | 1.64E-18 | 1 |
| 1 | 500 | 0.417 | 6.20E-16 | 1.28E-18 | 1 |
| 2 | 550 | 0.250 | 1.45E-15 | 1.90E-18 | 1 |
| 3 | 550 | 0.333 | 1.11E-15 | 1.64E-18 | 1 |
| 4 | 550 | 0.500 | 1.15E-15 | 1.51E-18 | 1 |
| 5 | 600 | 0.250 | 2.39E-15 | 2.21E-18 | 1 |
| 6 | 600 | 0.333 | 2.15E-15 | 2.20E-18 | 1 |
| 7 | 600 | 0.500 | 2.30E-15 | 2.62E-18 | 1 |
| 8 | 650 | 0.250 | 4.23E-15 | 3.29E-18 | 1 |
| 9 | 650 | 0.333 | 3.66E-15 | 3.21E-18 | 1 |
| 10 | 650 | 0.500 | 3.81E-15 | 3.02E-18 | 1 |
| 11 | 700 | 0.250 | 5.43E-15 | 3.41E-18 | 1 |
| 12 | 700 | 0.333 | 4.46E-15 | 3.56E-18 | 1 |
| 13 | 750 | 0.250 | 6.65E-15 | 2.86E-18 | 1 |
| 14 | 750 | 0.333 | 4.52E-15 | 4.01E-18 | 1 |
| 15 | 800 | 0.250 | 5.59E-15 | 3.13E-18 | 1 |
| 16 | 800 | 0.333 | 3.76E-15 | 2.88E-18 | 1 |
| 17 | 850 | 0.250 | 4.29E-15 | 2.93E-18 | 1 |
| 18 | 850 | 0.333 | 3.13E-15 | 2.72E-18 | 1 |
| 19 | 900 | 0.250 | 3.82E-15 | 3.08E-18 | 1 |
| 20 | 900 | 0.333 | 2.68E-15 | 3.01E-18 | 1 |
| 21 | 950 | 0.250 | 3.35E-15 | 2.68E-18 | 1 |
| 22 | 950 | 0.333 | 2.71E-15 | 2.66E-18 | 1 |

| Step Number | Temperature (°C) | Duration (Hours) | $^{39}$Ar Released (mol) | Measurement Uncertainty (mol; $1\sigma$) | Included in Diffusion Kinetics Optimization? (1=yes, 0=no) |
|---|---|---|---|---|---|
| 23 | 1000 | 0.250 | 3.54E-15 | 3.49E-18 | 1 |
| 24 | 1000 | 0.333 | 2.96E-15 | 2.67E-18 | 1 |
| 25 | 1050 | 0.250 | 4.02E-15 | 3.14E-18 | 1 |
| 26 | 1050 | 0.333 | 3.29E-15 | 2.90E-18 | 1 |
| 27 | 1100 | 0.250 | 4.75E-15 | 3.75E-18 | 1 |
| 28 | 1100 | 0.500 | 5.01E-15 | 2.75E-18 | 1 |
| 29 | 1100 | 1.000 | 5.95E-15 | 3.19E-18 | 1 |
| 30 | 1100 | 2.000 | 7.04E-15 | 3.86E-18 | 1 |
| 31 | 1100 | 3.333 | 6.73E-15 | 3.29E-18 | 1 |
| 32 | 1170 | 0.233 | 1.25E-14 | 5.57E-18 | 0 |
| 33 | 1185 | 0.233 | 1.20E-14 | 6.19E-18 | 0 |
| 34 | 1200 | 0.233 | 7.56E-15 | 4.37E-18 | 0 |
| 35 | 1215 | 0.233 | 2.91E-15 | 2.59E-18 | 0 |
| 36 | 1235 | 0.233 | 6.00E-16 | 1.31E-18 | 0 |
| 37 | 1280 | 0.233 | 6.11E-17 | 3.71E-19 | 0 |
| 38 | 1370 | 0.233 | 2.11E-17 | 2.49E-19 | 0 |

**Table A4.** Sample GR-27 input (Wong et al., 2023).

| Step Number | Temperature (°C) | Duration (Hours) | $^{39}$Ar Released (mol) | Measurement Uncertainty (mol; $1\sigma$) | Included in Diffusion Kinetics Optimization? (1=yes, 0=no) |
|---|---|---|---|---|---|
| 0 | 500 | 0.250 | 1.23E-15 | 2.19E-18 | 1 |
| 1 | 500 | 0.417 | 8.07E-16 | 1.55E-18 | 1 |
| 2 | 550 | 0.250 | 1.66E-15 | 2.25E-18 | 1 |
| 3 | 550 | 0.333 | 1.07E-15 | 1.56E-18 | 1 |
| 4 | 550 | 0.500 | 9.55E-16 | 1.46E-18 | 1 |
| 5 | 600 | 0.250 | 1.72E-15 | 1.91E-18 | 1 |
| 6 | 600 | 0.333 | 1.41E-15 | 1.92E-18 | 1 |
| 7 | 600 | 0.500 | 1.87E-15 | 1.86E-18 | 1 |
| 8 | 650 | 0.250 | 4.23E-15 | 3.52E-18 | 1 |
| 9 | 650 | 0.333 | 3.69E-15 | 2.82E-18 | 1 |
| 10 | 650 | 0.500 | 3.65E-15 | 3.31E-18 | 1 |
| 11 | 700 | 0.250 | 4.40E-15 | 3.73E-18 | 1 |
| 12 | 700 | 0.333 | 3.60E-15 | 2.76E-18 | 1 |
| 13 | 750 | 0.250 | 4.62E-15 | 3.53E-18 | 1 |
| 14 | 750 | 0.333 | 3.32E-15 | 2.79E-18 | 1 |
| 15 | 800 | 0.250 | 4.02E-15 | 3.26E-18 | 1 |
| 16 | 800 | 0.333 | 2.94E-15 | 2.38E-18 | 1 |
| 17 | 850 | 0.250 | 3.85E-15 | 2.81E-18 | 1 |
| 18 | 850 | 0.333 | 2.96E-15 | 2.35E-18 | 1 |
| 19 | 900 | 0.250 | 4.62E-15 | 3.34E-18 | 1 |
| 20 | 900 | 0.333 | 3.71E-15 | 2.78E-18 | 1 |
| 21 | 950 | 0.250 | 5.26E-15 | 3.46E-18 | 1 |
| 22 | 950 | 0.333 | 3.82E-15 | 4.29E-18 | 1 |

| Step Number | Temperature (°C) | Duration (Hours) | $^{39}$Ar Released (mol) | Measurement Uncertainty (mol; $1\sigma$) | Included in Diffusion Kinetics Optimization? (1=yes, 0=no) |
|---|---|---|---|---|---|
| 23 | 1000 | 0.250 | 4.68E-15 | 2.43E-18 | 1 |
| 24 | 1000 | 0.333 | 3.41E-15 | 2.96E-18 | 1 |
| 25 | 1050 | 0.250 | 4.61E-15 | 3.73E-18 | 1 |
| 26 | 1050 | 0.333 | 3.78E-15 | 3.05E-18 | 1 |
| 27 | 1100 | 0.250 | 6.47E-15 | 4.04E-18 | 1 |
| 28 | 1100 | 0.500 | 5.98E-15 | 3.31E-18 | 1 |
| 29 | 1100 | 1.000 | 6.76E-15 | 3.86E-18 | 1 |
| 30 | 1100 | 2.000 | 7.55E-15 | 4.27E-18 | 1 |
| 31 | 1100 | 3.333 | 6.95E-15 | 4.17E-18 | 1 |
| 32 | 1170 | 0.233 | 2.40E-14 | 9.26E-18 | 0 |
| 33 | 1185 | 0.233 | 1.78E-14 | 6.38E-18 | 0 |
| 34 | 1200 | 0.233 | 8.52E-15 | 5.10E-18 | 0 |
| 35 | 1215 | 0.233 | 2.82E-15 | 2.87E-18 | 0 |
| 36 | 1235 | 0.233 | 5.46E-16 | 1.04E-18 | 0 |
| 37 | 1280 | 0.233 | 4.93E-17 | 4.16E-19 | 0 |
| 38 | 1370 | 0.233 | 1.88E-17 | 2.55E-19 | 0 |

500