# Peer review of "An optimization tool for identifying Multiple Diffusion Domain Model parameters"

_Geochronology, 2024_

## Author Comment (AC1)

Andrew L. Gorin, MS
PhD Candidate
Department of Earth and Planetary Science
University of California, Berkeley
Berkeley, CA 94704
508-332-8101 · andrew_gorin@berkeley.edu

August 13, 2024

To the editor,

We thank the reviewers for their thoughtful comments. We are encouraged that they generally agree that the MDD toolkit approach is an improvement over past fitting methods and that our manuscript is good fit for Geochronology. To address these reviews, we intend to make the following changes:

- Additional context: As was constructively pointed out by both reviewers, we failed to appropriately describe many recent criticisms of the MDD Model. To address this comment, we will add a subsection where we review the history of the MDD model and the relevant criticisms. We will also comment on the extent to which these criticisms have been addressed. However, we note that it is out of the scope of our study to attempt to validate the MDD Model itself. Accordingly, we will adjust the tone of our language to ensure that we do not appear to be making claims to the contrary.

- Choice of Misfit statistic: Since both reviewers asked us to comment in more detail about the choice of misfit statistic, we will add additional language about this topic. Put simply, we do not believe we have a justification for the user to choose one over the other, and believe that when the two provide different answers, that the range of possibilities highlights the true uncertainty in the MDD model. We will elaborate on this view.

- Clarification of model application to Wong et al. (2023): Several specific points were made about the lack of perfect agreement between our new MDD model fits and the results from Wong et al. (2023). We will modify our language to emphasize three main points.

  First, we do not intend to suggest that our work provides a *better* fit than the one presented by Wong et al. (2023). Instead, we merely intend to suggest that our fits are, within the uncertainties of all the techniques presented (e.g. K-feldspar MDD, Biotite Ar, fission track, etc.), not excluded by pre-existing thermochronological data. Second, given that we have removed a user-defined model parameter—the $E_a$—we intend to emphasize that this general agreement is compelling. Third, Wong et al. (2023) explicitly states in their paper supplement that they considered a range of $E_a$ values and chose those in best agreement with independent thermochronological data. Any comparison of our model results with those of Wong et al. (2023) should be made with this in context in mind. In our revised manuscript, we will more clearly emphasize this point, as it was previously omitted.

- MDD model as physical reality: Several reviewer points are based on the assertion that the MDD model should map neatly to the physical structure of a given mineral. We will modify our language to more clearly articulate our position that the MDD model need not neatly map to the physical structure of a given mineral to be useful. While we do currently assert that it remains unclear what the model domains represent physically within a mineral, we further clarify our view on the non-physicality of the model.

The MDD model proposes that the diffusive behavior of certain minerals is best described by numerous, non-interacting, infinite sheets simultaneously diffusing within the same mineral. This description is self-evidently nonphysical. However, this non-physicality does not necessitate the rejection of the MDD model as a tool for deriving the thermal histories of minerals. Instead, it should be thought of as an empirical model. While the lack of physical description for the diffusive behavior of these minerals is indeed unsatisfying, evidence suggests that the MDD model reliably predicts thermal histories supported by independent thermochronological data. In this sense, the MDD model need not map neatly to reality to produce valuable insights.

- Line-by-line comments: The reviewers pointed out several typos, and suggestions for standardizing figure axes. We appreciate their thoroughness and will address these comments as suggested in our revised manuscript.

We appreciate the reviewers' time and their constructive comments. We look forward to the opportunity to revise our manuscript.

Best Regards,

Andrew L. Gorin

---

## Author Comment (AC2)

Andrew L. Gorin, MS
PhD Candidate
Department of Earth and Planetary Science
University of California, Berkeley
Berkeley, CA 94704
508-332-8101 · andrew_gorin@berkeley.edu

August 13, 2024

To the editor,

We thank the reviewers for their thoughtful comments and have revised our manuscript in accordance with their comments. We are encouraged that they generally agree the MDD_toolkit approach is an improvement over past fitting methods and that our manuscript is good fit for Geochronology.

Our most substantial revision to the manuscript was to add a discussion of critiques of the MDD model. We now review assumptions inherent to the MDD model as well as the ways they have been tested. Studies referenced by the community commenter were also incorporated.

Further revisions include rather minor clarifications about the methodology used by Wong et al., (2023). We now note that the models presented in Wong et al. (2023) were explicitly tuned to match independent thermochronological constraints. Because our approach removes this user intervention, one should not expect the thermal pathways generated by our MDD_toolkit to match these thermochronological constraints more closely than previously published models. We have carefully reviewed our language and now state only that our predicted thermal histories *also* agree with independent constraints.

The editor has asked that we provide a description of suggested revisions that we did not agree with. These comments are explained below with the original reviewer comment in blue and our response in black. We appreciate the reviewers' time and their constructive comments.

Best regards,

Andrew L. Gorin

**Reviewer 1.**

The cooling history for GR1 is initially pretty high temperature, especially when you compare it to the pre-existing BtAr date. The cooling history from Wong et al., 2023 would suggest that the sample experienced relatively monotonic cooling since BtAr closure. Is this realistic?

Given that GR1 is a part of a slowly cooled pluton, Wong et al.'s (2023) K-feldspar MDD prediction of monotonic cooling appears reasonable. That said, we do not find their prediction inconsistent with those made by our MDD-model framework. Looking first at the pathway predicted for GR1 from the $\%_{frac}$ misfit statistic, it is consistent with a monotonically-cooling system within the regions of error shown on Figure 4. While the history generated from the $\chi^2$ misfit statistic may appear to contradict the Biotite $^{40}Ar/^{39}Ar$ closure age the actual uncertainty on this age is arguably larger than shown in Figure 4.

Our plot shows the uncertainty bounds proposed by Wong (2023), but these may underestimate the true uncertainty for a few reasons. First, the plotted biotite-Ar closure temperatures from Wong (2023) are calculated assuming a cooling rate of 10°C/Ma, however, Wong et al.'s (2023, Figure 3) suggests that this cooling rate may have been as high as 60°C/Ma, which would increase the predicted closure temperature. Additionally, others have found biotite-Ar closure temperatures can vary significantly based on mineral chemistry and can be as high as high as 400 – 450 degrees (Hess et al., 1993, Grove 1996). Without detailed geochemical studies, one cannot rule out closure temperatures this high (Grove, 1996). Finally, the Wong et al. (2023) biotite-Ar age is older than the emplacement age of the pluton itself (Banks, 1972), consistent with this age potentially requiring a higher associated closure temperature than published by Wong (2023).

Wong et al. (2023) also explicitly tune their models to best match the existing thermochronological constraints. Because our new method avoids this user bias, it is unsurprising that the agreement appears less impressive. Overall, given the sources of uncertainty in both the biotite closure temperature and in the MDD model results, we do not find our predictions to be unrealistic.

**Reviewer 2.**

Line 95: Note that some iterations of the MDD model used variable $E_a$ see (*Lovera, O.M., Grove, M. and Harrison, T.M. 2002. Systematic analysis of K-feldspar Ar-40/Ar-39 step heating results II: Relevance of laboratory argon diffusion properties to nature. Geochimica Et Cosmochimica Acta, 66, 1237-1255.*)

After rereading the referenced study, we cannot find any instance of Lovera using variable $E_a$ in the referenced—or any other—study and therefore do not include references to this comment in our updated manuscript.

Line 171: Assuming the largest domain is similar to the size of the physical mineral grain the new approach (similar to earlier MDD models) indicates the smallest domains are commonly less than 1 micron. Given the known mineral and textural features of alkali feldspar of similar dimensions (e.g. Parson et al. 1999), please comment on the potential for 39Ar recoil affecting the estimates of the lower end of the output thermal history using the new approach.

The referenced line is describing how we created our synthetic dataset. We emphasize that this dataset is not a description of a real mineral and that our optimization should return the correct diffusion kinetics parameters regardless of whether the chosen kinetics might be considered "realistic".

That said, the point about alpha recoil has historically been raised in relation to MDD modeling (e.g. Harrison et al., 2014; Parsons et al., 1999; Villa et al., 1997). In our revised manuscript, we detail our view

that the MDD model is largely empirical, focusing on the unsatisfying lack of a physical manifestation for the domains. As such, we hesitate to speculate on the effects of alpha recoil on predicted thermal histories.